# Y Chromosome Loss Is a Frequent Event in Barrett’s Adenocarcinoma and Associated with Poor Outcome

**DOI:** 10.3390/cancers12071743

**Published:** 2020-06-30

**Authors:** Heike Loeser, Christina B. Wölwer, Hakan Alakus, Seung-Hun Chon, Thomas Zander, Reinhard Buettner, Axel M. Hillmer, Christiane J. Bruns, Wolfgang Schroeder, Florian Gebauer, Alexander Quaas

**Affiliations:** 1Institute of Pathology, University Hospital Cologne, 50937 Cologne, Germany; christina.woelwer@uk-koeln.de (C.B.W.); reinhard.buettner@uk-koeln.de (R.B.); axel.hillmer@uk-koeln.de (A.M.H.); alexander.quaas@uk-koeln.de (A.Q.); 2Department of General, Visceral and Cancer Surgery, University Hospital Cologne, 50937 Cologne, Germany; hakan.alakus@uk-koeln.de (H.A.); seung-hun.chon@uk-koeln.de (S.-H.C.); christiane.bruns@uk-koeln.de (C.J.B.); wolfgang.schroeder@uk-koeln.de (W.S.); florian.gebauer@uk-koeln.de (F.G.); 3Department of Internal Medicine I, University of Cologne, Center for Integrated Oncology Aachen Bonn Cologne Duesseldorf, 50937 Cologne, Germany; thomas.zander@uk-koeln.de

**Keywords:** Y chromosome loss, Barrett, esophageal adenocarcinoma, prognosis, tumor microenvironment

## Abstract

Background: The loss of the Y chromosome in various malignant diseases has been described previously. There are no reliable information on the actual frequency, significance and homogeneity of Y chromosome loss (*LoY*) in esophageal adenocarcinoma (EAC). Methods: 400 male EAC including lymph-node metastases were analyzed with commercially available Y chromosome specific fluorescence in-situ probes. The results were correlated with molecular and immunohistochemical markers and clinicopathological aspects. Results: The entire cohort (*n* = 400) showed a singular *LoY* of one chromosome arm in 1.0% (q-arm) and 2.8% (p-arm), complete *LoY* in 52.5%. *LoY* was strongly associated with shortened overall-survival (OS). Patients with preserved Y chromosome had a median OS of 58.8 months, patients with *LoY* an OS of 19.4 months (*p* < 0.001). Multivariate analysis showed *LoY* as an independent prognostic marker with a hazard ratio of 1.835 (95% CI 1.233–2.725). *LoY* correlated with *TP53* mutations (*p* = 0.003), *KRAS* amplification (*p* = 0.004), loss of ARID1a (*p* = 0.045) and presence of LAG3 (*p* = 0.018). Conclusions: Loss of the Y chromosome is a very common phenomenon in EAC. The *LoY* is heterogeneously distributed within the tumor, but corresponding lymph node metastases frequently show homogeneous *LoY,* indicating a selection and metastasizing advantage with poor prognosis. To date, the male predominance of EAC (7–9:1) is unclear, so genetic explanatory models are favored. The *LoY* in EAC may be biologically and functionally relevant and additional genomic or functional analyses are needed.

## 1. Introduction

The incidence of esophageal adenocarcinoma (EAC) has increased massively in recent decades, particularly in the Western world. Data from the United States show a 7-fold increase. In the past decades, there was an increase from 3.6 cases/1 million inhabitants in 1970s to 25.6/1 million inhabitants in 2006. A similar development is observed in Germany. The reason for this increase is unclear; a possible connection with rising obesity rates is discussed [1,2,3,4,5,6]. The vast majority of adenocarcinomas of the esophagus show a clear connection to the presence of a Barrett mucosa, which in turn is caused by chronic reflux of gastric acid and bile into the lower esophagus. While the majority of the studies describe a gender-neutral or up to twice as frequent occurrence of Barrett mucosa in men (maximum 2:1), the resulting adenocarcinoma shows an overwhelming male predominance of up to 9:1. The underlying reason for this gender-specific tumorigenesis is completely unclear and has not been investigated so far. A possible association with the androgen receptor signaling pathway has been discussed in the past with contradictory results [7,8,9]. In the past, the Y chromosome was presumed as limited to sex determination and spermatogenesis. Over the last years it has become evident that this disregard was false, as the Y chromosome is involved in many biological processes. An involvement is seen in coronary artery disease, including coronary artery disease, infectious diseases and, beyond hormonal influence, also autoimmune diseases [10,11,12,13,14]. Furthermore, the loss of the entire Y chromosome in various malignant diseases is a frequently described phenomenon. In bladder carcinoma, for example, loss of the Y chromosome is found in 30% of cases [15]. In older men, non tumor cells frequently show a loss of the Y chromosome - this is also discussed in the literature in connection with the overall higher propensity of the male sex for the development of a malignant disease [16]. Former studies considering only very small numbers of up to 20 cases of EAC have described the Y chromosome loss in up to 75% of the investigated cases [17,18,19,20,21]. 

To date, there is no reliable information on the actual frequency, significance for tumor progression including prognostic significance, the type of Y chromosome loss (isolated p-arm loss, isolated q-arm loss and complete Y chromosome loss) and the homogeneity of Y chromosome loss within the tumor in esophageal adenocarcinoma. Thus, we analyzed a very large tumor collective that includes 400 cases of EAC with different fluorescence in-situ probes and statistically analyzed them considering additional molecular tumor changes and clinical aspects. 

## 2. Results

### 2.1. Patient Samples and Correlation with Clinical Data

The entire cohort considered for analysis consisted of 400 male patients with EAC. Female patients were excluded by definition. From a subset of patients, lymph node metastases corresponding to the primary tumors were included in the TMA. Within the patients’ cohort, lymph node metastases of 255 patients were available, and from these, 165 cases were analyzable for Y chromosome loss (*LoY*) (64.7%). Neoadjuvant treatment (either chemo-radiation or chemotherapy) was administered in 223/400 patients (55.8%). The median follow-up for the entire cohort was 57.7 months with a calculated 5-year survival rate of 26.6%. The patient collective analyzed corresponds to the normal distribution in the frequency of the individual tumor stages. In the analyzed patient collective, tumors of all stages could be analyzed (pT1–pT4) with an accumulation of pT3/pN+ tumors (Table 1).

### 2.2. Y Chromosome Status in Cell Lines

Fluorescence in-situ hybridization (FISH) analysis of the cell lines fully agreed with the known Y chromosome status, thus the FISH probes used represent a reliable method for the analysis of the Y chromosome status (Figure 1).

### 2.3. Y Chromosome Status in the Patients’ Cohort

Singular loss of one of the Y chromosome arms was present in 0.1% (q-arm, *n* = 4) and 2.8% (p-arm, *n* = 11), complete *LoY* in 52.5% (*n* = 210) (Figure 2). *LoY* in lymph nodes was found with a higher frequency (60.6%) but with no statistical difference to the *LoY* frequency detectable in primary tumors (*p* = 0.125). *LoY* was not correlated to administration of any kind of neoadjuvant treatment, neither in primary tumors nor in lymph node metastasis. 

*LoY* was not associated with patients’ age (*p* = 0.124) or tumor stage (pT) (*p* = 0.314). In lymph node positive patients (pN+), *LoY* was seen in higher frequency 59.4%) than in nodal negative patients (*p* = 0.001) which is also reflected in an association with UICC stage and *LoY* (*p* = 0.015). 

### 2.4. Heterogeneity of the Y Chromosome Status

We analyzed ten large tumor slides evaluating the intra-tumoral heterogeneity of the Y chromosome via FISH-analysis. Therefore, we screened the whole tumor for the signal distribution revealing areas with preserved and lost signals of the Y chromosome. We found a heterogeneous *LoY* in five cases, some with only focal heterogeneity (Figure 2E,F). Furthermore, one of the ten cases had a complete *LoY*, whereas two cases showed a complete loss of one arm of the Y chromosome. Two cases had a homogeneously preserved Y chromosome.

### 2.5. Y Chromosome Status and Correlation with Immunohistochemical and Molecular Markers

Analysis of the molecular marker profile showed a correlation between LoY and TP53 mutations and KRAS amplifications (*p* = 0.003 and *p* = 0.004, respectively), loss of chromatin remodeling protein ARID1a (SMARCA4) (*p* = 0.045) and presence of the immune checkpoint regulator of LAG3 (*p* = 0.018). A correlation with amplification of Her2/neu, PIK3CA and GATA6 and CD3 positive T-cells could not be revealed (Table 1). 

### 2.6. LoY Correlation to Patients’ Outcome

*LoY* was strongly associated with shortened overall-survival (OS) in the entire patients´ cohort. Patients with presence of Y chromosome showed a median OS of 58.8 months (95% CI 33.1–83.2 months), patients with *LoY* an OS of 19.4 months (95% CI 14.8–24.0 months, *p* < 0.001) (Figure 3A). The survival difference is detectable both in patients that underwent upfront surgery without neoadjuvant treatment (median OS 117.7 months (95% CI 92.2–142.6 months) vs. 32.5 months (95% CI 13.1–51.9 months), *p* = 0.015) and in patients that received neoadjuvant treatment (median OS 41.3 months (95% CI 27.5–55.0 months) vs. 17.2 months (95% CI 12.6–22.0 months), *p* = 0.002) (Figure 3B,C).

Interestingly, patients with preserved Y chromosome in combination with high numbers of CD3 positive tumor infiltrating T-cells (CD3+ high) showed a significantly prolonged overall- survival compared to the group with *LoY* and low CD3+ status. This is reflected in the Kaplan-Meier survival analysis to the effect that in the group with preserved Y chromosome and CD3+ high the median OS is not reached, whereas in the group *LoY* and CD3+ low a median OS of 24.6 months (95% CI 19.4–30.1 months, *p* < 0.001) is seen. 

Next, we tested for independence of *LoY* as a prognostic marker using a multivariate cox regression model with age, tumor stage, lymph node metastases, and grading as covariates. *LoY* was seen as an independent prognostic marker with a hazard ratio of 1.835 (95%CI 1.835–2.725) (Table 2). 

## 3. Discussion

The extent of Y-chromosome loss in adenocarcinomas of the esophagus has so far only been determined on small case numbers. With 400 adenocarcinomas analyzed, this study is by far the largest tumor cohort investigated. Our study validated that the complete loss of the Y chromosome is a common phenomenon in EAC.

It cannot be ruled out that the *LoY* in the tumor is a biologically-functionally irrelevant epiphenomenon that occurs as a general expression of a “complex karyotype” of esophageal carcinoma. This may be supported by the fact that the EAC is often characterized by genomic chaos, including genome duplication, chromotrypsis or pronounced telomere shortening, which contribute to complex chromosomal rearrangements [22]. *TP53* mutations strongly correlate with instable genomes [23]. However, our multivariate regression analysis showed that *LoY* has a prognostic effect independent of *TP53* status indicating that genome instability with its high diversity of selective advantages might not be the (only) cause for *LoY* manifestation in EAC. Further, the extremely high frequency of >50% of *LoY* suggests that there is a fitness advantage. To our knowledge, EAC represents the highest rate of *LoY* among solid tumors [15,24,25,26]. Cestari et al. found *LoY* in high grade dysplastic Barrett’s mucosa but not yet in low grade dysplastic or normal Barrett’s mucosa. This could indicate that *LoY* is biologically relevant for tumor initiation [18]. Overall, it seems well possible that *LoY* in EAC has an oncogenic effect. The LoY is described to be associated with shorter survival and risk of cancer in general. Some studies even suggest to use the LoY in blood cells in healthy elderly men as a predictive biomarker for carcinogenesis [16]. 

*LoY* is frequently seen in circulating leukocytes, where it can be not only linked to a higher risk of hematological malignancies but indicates also a higher risk for the development of solid cancer [16,27]. Thompson et al. recently detected a relevant overlap of *LoY*-risk loci with known cancer susceptibility loci and somatic drivers of tumor growth, proposing a connection between *LoY* and genomic instability [27]. The mechanisms and implications of the LoY are not clear, so different hypotheses are discussed. Possible mechanisms of rising LoY rates in age could be telomere shortening or that the gonosomes replicate lately in the s-phase being more vulnerable for shortening of the cell cycle [28,29]. The *LoY* also seems to influence signal pathways as smoking has a transient and mutagenic effect on the Y chromosome status [30]. This matches the theory of altered tissue microenvironment signaling influencing somatic evolution in an age-dependent manner [31]. The LoY might affect survival through defective immune functions of blood cells by disrupting immunosurveillance enabling tumor development and expansion [16,30]. The main risk factor for the development of EAC is the Barrett mucosa. Barrett mucosa occurs up to twice as frequently in men as in women. The resulting EAC shows a male predominance of 7 up to 9 to 1 (in our tumor collective 9 to 1). The cause of this gender shift in tumor initiation is unclear. There is currently no convincing evidence that gender-specific lifestyle aspects can explain this difference, so that genetic explanatory models are favored. Among other things, a possible role of the androgen pathways or protective aspects of female sex hormones are discussed. As explained above, *LoY* is already found in high-grade dysplastic Barrett mucosa and thus at an early stage of tumor initiation. *LoY* might render its cancer-promoting effect through tumor suppressor genes located on the X chromosome that have no homologous copy in males and that might respond to *LoY* by epigenetic silencing. In women, some tumor suppressor genes on the X chromosome escape X-inactivation. One study has shown that six of such genes show increased loss-of-function mutations in male tumor diseases (ATRX, CNKSR2, DDX3X, KDM5C, KDM6A, and MAGEC3) [26]. For KDM5D encoded on the X-chromosome, a direct interaction with the androgen receptor in the tumor cell nucleus could be demonstrated, and a reduction in the expression of KDM5D leads to disruption of the androgen receptor pathway [32]. 

Alternatively, the so-called pseudo-autosomal region (PAR) of the sex chromosomes could be relevant for tumor initiation in EAC. In this case, *LoY* would be advantageous for tumor development for the same reasons as mentioned above [26,33,34,35,36,37,38]. 

Whether a similar functional interaction is important in the development of EAC tumors should be subject of future analyses. Since sex chromosomes are commonly excluded from copy number analyses of comprehensive next generation sequencing-based analyses, it is plausible that *LoY* has not been recognized as a relevant factor in large genomic studies on EAC [26]. 

Interestingly, here we found a statistically significant association between tumors with *LoY* and loss of expression of ARID1a, a member of the relevant SWI-SNF chromatin remodeling complex. This trend towards mutual exclusivity between *LoY* and SWI-SNF inactivation might indicate that *LoY* has an epigenetic effect that reduces the additional selective advantage of other epigenetic alterations on cancer development. 

Considering the phenomenon of *LoY* apparently correlating with LAG3 as a marker of the tumor microenvironment in EAC could also implicate a direct influence of Y deficient leukocytes on the tumor immune microenvironment in cancer. Former studies linked the Y chromosome to infectious and autoimmune diseases indicating direct influence of the Y chromosome on the immune system [10,13,14,16]. For example, the development of NK-T-cells is directly associated with Y chromosome-linked factors [11]. Our present study has few limitations, as we performed no analysis of the Y chromosome status of blood cells. Such analyses, however, might help enlightening the link between the Y chromosome and the immune system in cancer. *LoY* is known to be the most common acquired mutation in healthy males with a rising prevalence with age and is found in up to 20% of men older than 80 years [39]. For colorectal, prostate, bladder and lung cancer studies showed a significant higher frequency of *LoY* in blood cells compared to healthy controls [40,41]. A further limitation is that the high frequency of *LoY* in fully developed EAC does not explain the gender specificity of the tumor, as the Y chromosome status in Barrett´s mucosa without dysplasia has not been analyzed in large cohorts yet [20,42,43]. 

## 4. Materials and Methods 

### 4.1. Patients and Tumor Samples

We analyzed formalin-fixed, paraffin embedded (FFPE) material from 400 male patients with EAC who underwent primary surgical resection or resection after neoadjuvant therapy at the Department of General, Visceral and Cancer Surgery, University of Cologne, Cologne, Germany. The standard surgical procedure was laparotomic or laparoscopic gastrolysis and right transthoracic en bloc esophagectomy including two-field lymphadenectomy of mediastinal and abdominal lymph nodes. Reconstruction was performed by high intrathoracic esophagogastrostomy as described previously [44,45]. Patients with advanced esophageal cancer (cT3, cNx, M0 or cN+, M0) received preoperative chemoradiation (5-FU, cisplatin, 40 Gy as treated in the area prior the CROSS trial) or chemotherapy alone. All patients were followed up according to a standardized protocol. During the first 2 years, patients were followed up clinically in the hospital every 3 months. Afterwards, annual exams were carried out. Follow-up examinations included a detailed history, clinical evaluation, abdominal ultrasound, chest X-ray and additional diagnostic procedures as required. Follow-up data were available for all patients. Patient characteristics are given in Table 1. Depending on the effect of neoadjuvant chemo- or radiochemotherapy, there is a preponderance of minor responders in the TMAs, defined as histopathological residual tumour of ≥10% [46].

All procedures followed the national and institutional ethical standards and were in accordance with the relevant version of the Helsinki Declaration. Informed and ethical approved consent from the local ethics committee (13-091) was obtained from all included patients.

### 4.2. Cell Culture

Commercially available FISH probes were tested and validated in different esophageal cell lines of known Y chromosome status. Immortalized human normal esophageal squamous epithelium cell lines EPC1-hTERT (XY) and EPC2-hTERT (XY) were gifts by Dr. René Thieme (Gockel laboratory, Leipzig). EPC-1 and EPC-2 cells were cultivated in Keratinocyte-SFM (KSFM) medium (Gibco, ThermoFisher Scientific, Waltham, MA, USA) supplemented with bovine pituitary extract (50 μg/mL) (Gibco, ThermoFisher Scientific, Waltham, MA, USA), and human recombinant epidermal growth factor (EGF) (1 ng/mL) (Gibco, ThermoFisher Scientific, Waltham, MA, USA). Following dissociation during subculturing, trypsin activity was blocked by soybean trypsin inhibitor (STI) (Sigma-Aldrich, Schnelldorf, Germany). The metaplastic cell line CP-A (XY) and the dysplastic cell line CP-B (XY) were purchased from ATCC and cultivated according to manufacturer’s recommendations. Esophageal adenocarcinoma cell lines OE-33 (X0), OE-19 (X0), Eso-26 (XY) and OAC-P4C (X0) were available from the German Collection of Microorganisms and Cell Cultures GmbH (DSMZ, Leibnitz Institute, Braunschweig, Germany). All EAC cell lines were cultivated in RPMI 1640 GlutaMax growth medium (Gibco, ThermoFisher Scientific, Waltham, MA, USA), supplemented with 10% fetal bovine serum (FBS). All cells were incubated at 37 °C and 5% CO_2_.

### 4.3. Formalin-Fixation and Paraffin-Embedding of Cell Line Pellets

Cells of 80% confluency were harvested by trypsinization (0.25% Trypsin + EDTA) (Gibco, ThermoFisher Scientific, Waltham, MA, USA), washed and resuspended in 4% paraformaldehyde (Walter-CMP, Ratingen, Germany). Cells were fixed overnight at 4 °C. Fixed cells were washed at room temperature (1200 rcf, 15 min) and resuspended in EtOH (97%). After adding three drops of protein glycerol (Carl Roth, Karlsruhe, Germany), cells were vortexed and subsequently spun (1200 rcf, 15 min). The cell pellet was dehydrated and embedded in paraffin.

### 4.4. TMA Construction

TMA construction was performed as previously described [47,48]. In brief, tissue cylinders with a diameter of 1.2 mm each were punched from selected tumor tissue blocks using a self-constructed semiautomated precision instrument and embedded in empty recipient paraffin blocks. Consecutive sections of the resulting TMA blocks were transferred to an adhesive-coated slide system (Instrumedics Inc., Hackensack, NJ) for immunohistochemistry and FISH.

### 4.5. Fluorescence In Situ Hybridization

FISH analysis of Y chromosome was performed with a panel containing probes for long and short arms of the Y chromosome (Abbott Molecular, Wiesbaden, Germany). Detailed procedures of FISH analysis were described elsewhere [49]. In brief, three-μm tissue sections (SuperFrost Plus) were mounted by heating at 56 °C, followed by semi-automated deparaffinization protease digestion washing steps (VP2000 processor system, Abbott Molecular, Germany) with the ready-to-use FISH pretreatment kit (Vysis IntelliFISH Universal FFPE Tissue Pretreatment Protease; Abbott Molecular, Wiesbaden, Germany). Hybridization at 37°C was done overnight with the FISH probes, followed by DAPI staining. Non tumor epithelial tissue and fibroblasts served as on-slide internal positive control except for the analysis of the cell lines, where no internal control was available by definition. Further analysis of the Y chromosome for the TMA and large tumor slides was only performed with internal positive control exhibiting clearly distinct signals of each color. The tumor spots were screened for existence or absence of green (long arm; Yq12) and orange (short arm; Yp11.3) signals. Complete loss of both was defined as loss of the Y chromosome, whereas loss in some tumor cells was defined as a mosaic pattern. Loss of one color (green or orange) was defined as partial Y chromosome loss. 

Further FISH analyses were performed on TMA slides for gene amplification of *KRAS*, *PIK3CA*, *Her2/neu (ERBB2)* and *GATA6*. A detailed description of the analysis of *KRAS*, *PIK3CA*, *Her2/neu* and *GATA6* is already published [50,51,52]. 

### 4.6. Analysis of Heterogeneity of Y Chromosome Status

To address the question of the distribution of the Y chromosome within the tumor we further performed FISH analyses on 10 large tumor slides evaluating the heterogeneity of the Y chromosome status. The analysis itself was done analogously to the TMAs. We chose cases where we observed differences of Y chromosome in the tumor and the lymph node metastasis on the TMA. We screened the whole tumor for the signal distribution of the short and long arm of the Y chromosome revealing areas with preserved and lost signals. 

### 4.7. Immunohistochemistry

Immunohistochemistry was performed on TMA slides evaluating the expression of CD3, TP53, LAG3, ARID1a and BRG1. A detailed description of the analysis for CD3, TP53, ARID1a and BRG1 is published [45,53]. For CD3 the rabbit monoclonal antibody (SP7; Thermo Fisher Scientific, MA, USA) and for LAG3 the rabbit monoclonal antibody for LAG3 (D2G40; Cell Signaling Technology Europe) was used on the Leica BOND-MAX Stainer (Leica Biosystems, Wetzlar, Germany). For CD3, <50 positive T-lymphocytes/mm^2^ were defined as low positive, >50 as highly positive, and for LAG3 expression in <1% lymphocytes were defined as negative and ≥1% was assessed as positive.

### 4.8. Data Analysis and Statistics

The current retrospective study was performed with the approval of the Ethics Committee of the University of Cologne, utilizing clinical data that was collected prospectively according to a standardized protocol.

Univariate analysis was conducted for tables using chi-squared statistics or Fisher’s exact test if necessary. Prognosis was calculated including all types of mortality beginning on the date of surgery. Kaplan–Meier univariate analysis was used to describe survival distribution, and log-rank tests were used to evaluate survival differences. Cox proportional hazard regression with sequential backward elimination of the non-significant variables was used to analyze the effect of several risk factors on survival. Survival analysis and the multivariate cox-regression model was performed on the entire cohort (test + validation cohort). 

Statistical analyses were carried out using IBM SPSS v22.0 (IBM Corporation, Armonk, NY, USA). 

## 5. Conclusions

Taken together, here we validated the high frequency of *LoY* in a large cohort of EAC. EAC patients with *LoY* have a poor prognosis and show correlation with the immune checkpoint protein LAG3, suggesting a direct influence of Y deficiency on the tumor immune microenvironment. Further, we observe a correlation between *LoY* and inactivation of SWI/SNF components raising the possibility of an epigenetic effect of *LoY*. Our data suggest that *LoY* has to be seen in a larger functional context. Understanding this context may shed light on the gender bias of EAC incidence. 

## Figures and Tables

**Figure 1 cancers-12-01743-f001:**
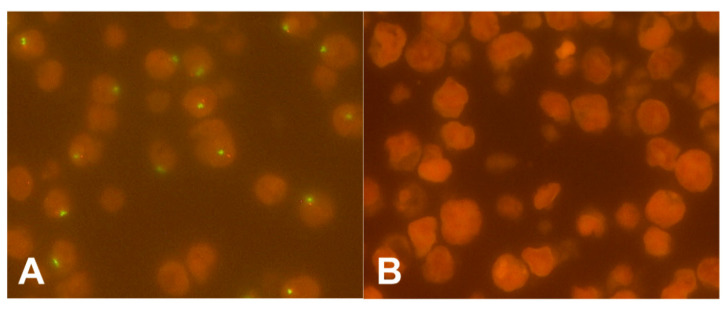
FISH of esophageal adenocarcinoma cell lines: (**A**) Preserved Y chromosome signals of the short (red signal) and long (green signal) arm; (**B**) Complete loss of the Y chromosome (magnification ×630).

**Figure 2 cancers-12-01743-f002:**
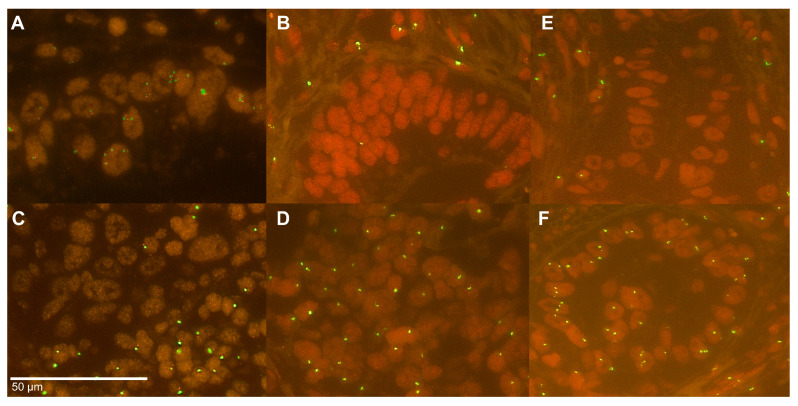
FISH of the Tissue Microarray: (**A**) preserved Y chromosome signals of the short (red signal) and long (green signal) arm; (**B**,**C**) complete loss of the Y chromosome; (**D**) mosaic pattern of the Y chromosome with partial loss (only long arm with green signals); (**E**,**F**) heterogeneity of the Y chromosome with partly lost and preserved signals in the tumor cells (magnification ×630).

**Figure 3 cancers-12-01743-f003:**
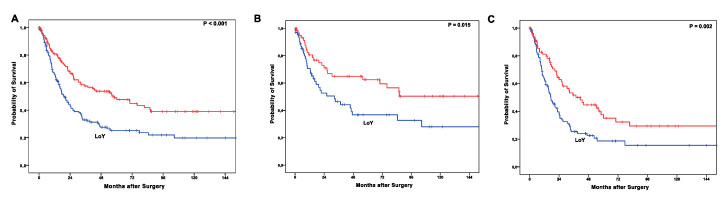
Kaplan-Meier survival analysis showing shortened overall survival (OS) in case of LoY (blue line) vs. preserved Y chromosome (red line) in the entire patients’ cohort (**A**). The effect of impaired OS is detectable in both, patients after upfront surgery without neoadjuvant treatment (**B**) and in patients who received neoadjuvant treatment (**C**).

**Table 1 cancers-12-01743-t001:** Patients’ characteristics and clinicopathological data; results of immunohistochemical and molecular markers considering the Y chromosome status.

Cliniopathological Parameter	Y Chromosome							
	Total			Loss		Presence		*p* Value
age group	<65 yrs	210	52.8%	117	56.0%	93	44.0%	
	>65 yrs	190	47.2%	91	47.9%	99	52.1%	0.124
tumor stage	pT1/2	109	27.4%	52	48.6%	56	51.4%	
	pT3/4	289	72.6%	157	54.3%	132	45.7%	0.314
lymph node metastasis	pN0	159	39.9%	68	42.8%	91	57.2%	
	pN+	239	60.1%	142	59.4%	97	40.6%	0.001
grading	G1	4	1.4%	0	0.0%	4	100.0%	
	G2	164	56.0%	70	48.5%	85	51.5%	
	G3	123	42.3%	73	59.3%	50	40.7%	0.031
neoadjuvant treatment	no	177	44.3%	87	49.2%	90	50.8%	
	yes	223	55.7%	123	55.2%	100	44.8%	0.268
*TP53*	wildtype	122	41.5%	53	43.4%	69	56.6%	
	mutation	172	58.5%	106	61.6%	66	38.4%	0.003
*KRAS*	negative	318	81.3%	159	50.0%	159	50.0%	
	positive	73	18.7%	50	68.5%	23	31.5%	0.004
*HER2/neu*	negative	247	87.0%	130	52.6%	117	47.4%	
	positive	37	13.0%	22	59.5%	15	40.5%	0.483
ARID1a	loss	40	10.4%	15	37.5%	25	62.5%	
	presence	343	89.6%	188	54.8%	155	45.2%	0.045
BRG1	loss	16	4.1%	5	31.3%	11	68.8%	
	presence	371	95.9%	202	54.4%	169	45.6%	0.078
*GATA6*	negative	340	89.5%	175	51.5%	165	48.5%	
	positive	40	10.5%	23	57.5%	17	42.5%	0.507
*PIK3CA*	negative	335	94.1%	181	54.0%	154	46.0%	
	positive	21	5.9%	8	38.1%	13	61.9%	0.180
CD3 infiltrating cells	low	203	70.2%	112	55.2%	91	44.8%	0.699
	high	86	29.8%	45	52.3%	41	47.7%	
LAG3	negative	290	93.9%	159	54.8%	131	45.2%	
	positive	19	6.1%	5	26.3%	14	73.7%	0.018

**Table 2 cancers-12-01743-t002:** Multivariate cox regression demonstrating *LoY* as an independent prognostic marker for overall survival.

Clinicopathological Parameter		Hazard Ratio	95% Confidence Interval		*p* Value
			lower	upper	
age group	<65 yrs. vs. >65 yrs	1.435	0.978	2.105	0.065
tumor stage	pT1/2 vs. pT3/4	1.195	0.704	2.027	0.51
lymph node metastasis	pN0 vs. pN+	2.877	1.839	4.502	<0.001
grading	G1 vs. G2/3	1.572	1.082	2.284	0018
*TP53*	wildtype vs. mutation	1.120	0.756	1.658	0.573
*LoY*	presence vs. loss	1.835	2.725	1.233	0.003

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
