# Peer review of "Y Chromosome Loss Is a Frequent Event in Barrett’s Adenocarcinoma and Associated with Poor Outcome"

_cancers, 2020, doi:10.3390/cancers12071743_

Round 1

Reviewer 1 Report

This study shows that loss of Y chromosome (LoY) frequently occurs in esophageal adenocarcinoma (EAC), is associated with a reduced overall survival, and may provide additional prognostic ability over and above the currently known prognostic markers. The findings of this study are interesting and significant in terms of EAC genetics. However, I have several concerns:  

Although this study shows that LoY is associated with EAC survival, the results may not necessarily show that it is also associated with EAC risk and incidence. A comparison of frequency of LoY in age-matched non-tumor controls and Barrett’s non-tumor tissues are required to confirm an association and its magnitude of effect on EAC risk. For the same reason, although suggestive, these results may not necessarily show why EAC is more common in males than in females (as its association with EAC risk remains unclear). I suggest the authors investigate or discuss these points as limitations of this study.

In the multivariate cox regression analysis, age, tumor stage, lymph node metastases, and grading were used as covariates. Did the author also include the other variables that were associated with LoY in Table 1 (e.g. KRAS, etc)? Or were those variables excluded because they were non-significant in the backward Cox proportional hazard regression?

The results of this study could suggest that LoY may also be associated with tumor progression over time. However, a more detailed analysis on this requires time-to-event data for progression from one stage to another. Do the author have sufficient data and statistical power to investigate this?

In the Kaplan-Meier survival analysis, the overall survival was reduced in patients who received neoadjuvant treatment. Is this influenced by the severity or grading of the tumors?

In the sentence “Survival analysis and the multivariate cox-regression model was performed on the entire cohort (test + validation cohort)”, I was not clear what the validation and test cohorts were referring to. Was there a validation cohort involved in this study?

In the multivariate survival analysis, it appears that LoY affects survival, partly independent of tumor severity and grading. It might be helpful to discuss through what alternative mechanisms LoY could affect survival (e.g. metastasis, etc)?

For the sentence, “LoY was not correlated to administration of any kind of neoadjuvant treatment, neither in primary tumors nor in lymph node metastasis”, the statistics for the stratified analyses (neoadjuvant treatment in primary tumors vs lymph node metastasis) has not been provided.

The section related to the study of the heterogeneity of Y chromosome status was not detailed. Please provide further details such as how heterogeneity was detected? Maybe presenting figures from histology slides can help?

There are some inconsistencies between the numbers provided throughout the text and Table 1. For example, in the main text, we read: “Within the patients’ cohort, lymph node metastases of 255 patients were available, and from these 165 cases were analyzable for LoY (64.7%).” However, these numbers do not match with those presented in Table 1. In addition, some of the numbers in Table 1 do not add up. For example, the total N is 189 for the age group > 65 while the numbers for the Y chromosome loss (N=103) and presence (N=96) groups sum up to 199.  

The authors mention “Interestingly, here we found a statistically significant association between tumors with LoY and loss of expression of ARID1a”. However, the association seems to be in the opposite direction as tumors with loss of ARID1a have a less frequent LoY in Table 1?

Some sentences are hard to follow, and may need to be re-written for a better clarity:

Example 1: “The patient collective analyzed corresponds to the normal distribution in the frequency of the individual tumor stages”. Does this sentence say that the frequency of the stages of tumors were normally distributed? But frequency of tumor stages is not a quantitative trait?

Example 2: “In the analyzed patient collective tumors of all stages could be analyzed (pT1 - pT4) with an accumulation of pT3/pN+ tumors (Table 1).”

Example 3:  “Interestingly, patients with presence of the Y chromosome in combination with high numbers of CD3 positive tumor infiltrating T-cells revealed long-term survival resulting in a median OS that was not reached in the Kaplan-Meier survival analysis compared to a median OS of 24.6 months (95% CI 19.4 – 30.1 months, p < 0.001) in the group with Y chromosome in combination with low number of T- cells in the tumor microenvironment”. Does this sentence say that survival in those groups were not influenced by LoY?

Figure 2 - is there a missing part in the Figure legend?

Reviewer 2 Report

The title reflects the subject of the study. This manuscript however does not present a clear and clinically useful message. It is not well written in terms of clarity, style, and use of English language. Materials and methods aren't sufficiently detailed. The discussion section doesn't explain the purpose of this study adequately. The length of the manuscript is good. All references are appropriate and current. I suggest that it must be rejected without revision.

Author Response

We are sorry to hear that you didn´t find the manuscript suitable for publication. We improved the manuscript according to the suggestions of the other Reviewers considering especially the Discussion section with amendments in the Material and Methods as well as Results section. If there are special further issues to be addressed, we will change them too.

Reviewer 3 Report

In this study, Loeser et al. have investiaged Y Chromosome Loss as a prognostic marker in male patients with Barrett´s adenocarcinoma. First, they have investigated and validated their assay by FISH analysis in well-selected EAC cell lines. In the next approach, the authors have evaluated the heterogeneity of the Y chromosome status in ten large tumor slides. This analysis demonstrates that the loss of the Y chromosome (LoY) can be very heterogenous. Susbsequently, they have evaluated the LoY in a large and very-well characterized cohort of 400 patients with EAC. The investigators can show that the loss of the Y chromosome is associated with a dismal prognosis. In a further subgroup analysis, they confirm their findings for paitents with upfront surgery and patients that have received a neoadjuvant chemotherapy. Eventually, the prognostic impact of LoY is confirmed by a multivariate analysis, showing still high significance and reveals LoY as an independent prognostic marker..

Overall, I have a very positive impression of the study. In contrast to other IHC biomarker, LoY seems to be a very robust surrogate marker which is less biased by an individual evaluator.. Moreover, so far, there exist only a few data for this biomarker. Moreover, the further clinical and molecular biological data of this cohort are also very comprehensive, which further emphasizes the role of the biomarker described here.

I have only some minor comments to make:

  • Is there also a prognostic role for the loss of the Y chromosome for recurrence free survival?
  • Does the Y chromose status correlate with an organotropic metastasis pattern?

Reviewer 4 Report

The authors in this manuscript entitled “Y chromosome loss is a frequent event in battrett’s adenocarcinoma and associated with poor outcome” examined the state of Y chromosome alterations in 400 human male esophageal adenocarcinoma (EAC) specimens by using immunostaining and then analyzed the correlation between these alterations and EAC stages, grading, metastasis, patient’s ages, altered tumor suppressor/oncogenes, and patient overall survival (OS). The information was previously unknown. The analyses of Y chromosome partial and complete loss as well as q-arm and p-arm Y chromosome alterations showed that Y chromosome loss is a common phenotype among AEC; AEC patients with Y chromosome loss are associated with reduced OS; and Y chromosome loss is correlated with increased TP53 mutations and KRAS amplification; as well as Y chromosome loss is associated with tumor’s grading and metastasis. The study is interesting and provides the new information.

A Concern:

In Figure 3, the labels (writing) for A, B, and C panels are too small to read. They authors should improve the Figure 3.

Round 2

Reviewer 2 Report

All reviewers' comments have been addressed. This manuscript presents a clear and clinically useful message. It is well written in terms of clarity, style, and use of English language. Materials and methods are sufficiently detailed. The discussion section explains adequately the purpose of this study in the context of published information. The conclusions accurately and clearly explain the main results. The length of the manuscript is ideal. All figures are of good quality and relevant to the subject. All references are appropriate and current.